# An Efficient and Conditional Privacy-Preserving Heterogeneous Signcryption Scheme for the Internet of Drones

**DOI:** 10.3390/s23031063

**Published:** 2023-01-17

**Authors:** Muhammad Asghar Khan, Insaf Ullah, Ako Muhammad Abdullah, Syed Agha Hassnain Mohsan, Fazal Noor

**Affiliations:** 1Department of Electrical Engineering, Hamdard University, Islamabad 44000, Pakistan; 2Computer Science Department, College of Basic Education, University of Sulaimani, Sulaimaniyah 00964, Kurdistan Region, Iraq; 3Department of Information Technology, University College of Goizha, Sulaimaniyah 00964, Kurdistan Region, Iraq; 4Ocean College, Zhejiang University, Zheda Road 1, Zhoushan 316021, China; 5Faculty of Computer and Information Systems, Islamic University of Madinah, Madinah 400411, Saudi Arabia

**Keywords:** Internet of Drones, security, privacy, signcryption, HECC

## Abstract

The Internet of Drones (IoD) is a network for drones that utilizes the existing Internet of Things (IoT) infrastructure to facilitate mission fulfilment through real-time data transfer and navigation services. IoD deployments, on the other hand, are often conducted in public wireless settings, which raises serious security and privacy concerns. A key source of these security and privacy concerns is the fact that drones often connect with one another through an unprotected wireless channel. Second, limits on the central processing unit (CPU), sensor, storage, and battery capacity make the execution of complicated cryptographic methods onboard a drone impossible. Signcryption is a promising method for overcoming these computational and security limitations. Additionally, in an IoD setting, drones and the ground station (GS) may employ various cryptosystems in a particular region. In this article, we offer a heterogeneous signcryption scheme with a conditional privacy-preservation option. In the proposed scheme, identity-based cryptography (IBC) was used by drones, while the public key infrastructure (PKI) belonged to the GS. The proposed scheme was constructed by using the hyperelliptic curve cryptosystem (HECC), and its security robustness was evaluated using the random oracle model (ROM). In addition, the proposed scheme was compared to the relevant existing schemes in terms of computation and communication costs. The results indicated that the proposed scheme was both efficient and secure, thereby proving its feasibility.

## 1. Introduction

The term “Internet of Drones” (IoD) refers to a network for interconnected drones and a ground station (GS) that allows the drones to enter low-altitude controlled airspace in a coordinated fashion. Drones in IoD networks typically have their sensors, software, and the technologies that connect them configured so that they may interact over the Internet using the same standard IoT protocols as other connected devices [1]. Historically, drones have been exploited for a large number of military applications and activities. However, due to substantial improvements in the design and manufacturing of inexpensive, highly reliable, and small-sized drones, drones are now being employed in a large array of civil and commercial applications. Moreover, the unique attributes of drones such as their ease of use, fast deployment to remote locations, high mobility, maneuverability, and capability to hover make them a suitable choice for commercial applications [2]. Despite their various benefits, there are still obstacles to overcome before IoD networks can be deployed successfully. Drones in IoD networks, for example, communicate through an unencrypted wireless channel; hence, it is essential to employ a cryptographic method with the highest level of security to enable their safe deployment in mission-critical situations [3]. Drones have limited onboard components such as CPUs, sensors, storage, and batteries [4]. Due to their small size, drones can only carry a limited number of supplies. Drones were designed for aerial surveillance with the primary goal of collecting data for transmission to the GS. Since drones often have small amounts of onboard storage and processing power, it can be difficult for them to perform complex computations. These restrictions may have a major impact on the privacy and security aspects of the IoD networks, which could lead to a catastrophic failure of the network’s information-exchange capacity [5].

In the absence of countermeasures against cyber-physical threats to preserve data security and privacy in IoD networks, it is possible for intruders to penetrate the network and disclose sensitive data. Examples of common privacy and security threats in the IoD ecosystem include drone position tracking, device tampering, unauthorized data access, message manipulation, and falsification. Global Positioning System (GPS) spoofing attacks [6,7,8] generally exploit GPS signals and pose a significant threat to the privacy of IoDs. By sending significantly more powerful fake GPS signals to a drone, an attacker can trick it into flying in the wrong direction during a GPS spoofing attack. Data integrity and confidentiality can be jeopardized when malicious actors introduce chaos into a network and steal sensitive information. To maximize the use of drones, it is vital to protect IoD networks with stronger security measures and a cryptographic algorithm that requires less computation.

The IoD must assure authenticity and confidentiality for it to be of the utmost importance. The digital signature and encryption methods address these security attributes respectively. When the need arises for both encryption and digital signatures, signcryption [9] can be employed. Due to the growing variety and density of drones, a given zone may contain drones and GSs that belong to different cryptosystems. Furthermore, drones have limited computational capacity and storage space. Consequently, an efficient and secure heterogeneous signcryption scheme in which the sender and recipient have independent security domains is a better option [10,11]. Consequently, identity-based cryptography (IBC) [12] and public key infrastructure (PKI) are the two main cryptosystems that can be implemented in the IoD system. In addition to a heterogeneous signcryption scheme, a conditional privacy-preservation feature can be introduced to ensure receiver and sender identity anonymity [13]. To prevent their real identity from being revealed to the sender and the receiver, each entity in the proposed scheme encrypts its identity using a secret key known only to the entity and the PKG throughout the key-generation process. In order to decipher the identification after the PKG has received it, it must first find the secret key and the real identity. The PKG then makes available the encrypted identities of all entities via signcryption and unsigncryption processing.

Typically, Rivest–Shamir–Adleman (RSA), bilinear pairing (BP), and elliptic curve cryptography (ECC) are employed to increase the security and efficiency of any security solution. RSA is based on a massive factorization problem and employs 1024-bit keys, parameters, certificates, and identities. RSA is inappropriate for resource-constrained networks such as IoD due to the lack of onboard processing capability on small drones. In addition, BP is inferior to RSA due to its extensive pairing and map-to-point function processing. ECC was developed to address the shortcomings of RSA and bilinear pairing. ECC typically uses 160-bit keys, which are again not suitable for IoD networks. Hyperelliptic curve cryptography (HECC), which is an improved variant of ECC, was developed to compete with ECC’s efficiency [14]. HECC offers the same amount of security as ECC, BP, and RSA with 80-bit keys. Therefore, HECC is the best choice for IoD systems, so we used it to construct the proposed scheme with the following main contributions.

We proposed a heterogeneous signcryption scheme in which the drone side utilized IBC and the GS side used PKI. The real identity of each entity was encrypted using a secret key that only the entity and the PKG knew during the key-generation process. This made the proposed scheme conditionally privacy-preserving.In the proposed scheme, we introduced a new concept in IBC in which the PKGC sent the private key to drones in an encrypted format that did not require a secure channel. Moreover, the proposed scheme was constructed using the concept of the HECC and assessed using a random oracle model (ROM). The results verified that the proposed scheme was robust against cyberattacks.Finally, we conducted a comparison study to evaluate the efficiency of the proposed scheme in terms of computation and communication costs. Comparing the proposed scheme to similar existing ones revealed that it had reduced computation and communication costs.

This manuscript is structured in a manner that includes the following sections: the related work on conditional privacy-preserving heterogeneous signcryption schemes is covered in Section 2, and the preliminary material is discussed in Section 3. In Section 4, we cover the construction of the proposed scheme. Security models are discussed in Section 5, and Section 6 provides a security analysis of the proposed scheme. We cover the performance analysis in Section 7, and the conclusions are contained in Section 8.

## 2. Related Work

Recent advancements in 5G technology have allowed the development of B5G cellular networks, which enable autonomous drone services. However, issues regarding the security and privacy of drones have increased rapidly [15]. The IoD’s wireless communications can be attacked in a number of ways using cryptographic techniques [16]. Therefore, an efficient and highly secure cryptographic scheme is required for the successful deployment of IoD networks. Sign-then-encrypt approaches meet network security standards; however, this strategy raises computation costs on both ends. One way to address this issue is signcryption, a sophisticated method that combines a digital signature and encryption in an operation that conducts them simultaneously. This method, which is both effective and well suited for devices with limited resources, is in contrast to the more standard practice of employing separate procedures for encryption and digital signatures [17]. Most existing signcryption solutions rely on PKI and IDC cryptosystems. However, these cryptosystems can only be functional in networks in which both the senders and receivers employ the same cryptographic mechanism for exchanging data. Heterogeneous signcryption is preferable due to the dynamic nature of IoD systems [18].

The first heterogeneous signcryption scheme between PKI and IBC was introduced by Sun and Li [19]. Huang et al. [20] highlighted the security shortcomings of [19] and offered a more robust security approach that was termed “insider security” before proposing a new scheme between PKI and IBC. Their schemes, however, did not enable batch unsigncryption. Ali et al. [21] developed a conditional privacy-preserving hybrid signcryption scheme that combined BP with heterogeneous communication. The protocol ensured that a message sent via the IBC method was delivered via a PKI method. Unfortunately, in this design, any entity was able to produce a pseudo-identity and a public key, whilst the recipient had no method to check its authenticity. In addition, their scheme failed to ensure inner unforgeability because a hostile receiver could easily intercept a valid ciphertext, produce a new random number, and forge a new valid ciphertext. Furthermore, the proposed method employed bilinear pairing, which is a costly process for drones to execute. Elkhalil et al. [22] developed an efficient signcryption of a heterogeneous system to offer high-level security properties such as confidentiality, key revocation, integrity, authentication, and nonrepudiation. The proposed scheme was based on ECC, a procedure that is slightly more expensive than HECC.

Jin et al. [23] presented a signcryption scheme that was provably secure and heterogenous for a smart grid system in which meters in the IBC environment communicated data to utilities in the PKI environment. The signcryption and unsigncryption algorithms in their method were computationally and communicatively inefficient due to the BP operations. In addition, the scheme did not support the decryption of numerous ciphertexts in bulk. Ting et al. [24] proposed an efficient online/offline heterogeneous signcryption scheme that met the security objectives of confidentiality, integrity, authentication, and nonrepudiation in a single logical step. In particular, its structure enabled a sensor node in an IBC configuration to send a message to an Internet host in a PKI, thereby reducing the rigorous verification demands on low-power devices. However, the proposed method was computationally expensive due to the ECC operation, which is difficult for a drone to execute. Ali et al. [25] introduced a hybrid signcryption technique that satisfied the security requirements for heterogeneous vehicle-to-infrastructure (V2I) communications in a single logical step. The scheme permitted the secure communication of safety messages from a vehicle to a roadside device using PKI. The basis of the proposed solution was ECC, which incurred lower communication and computation costs. Pan et al. [26] presented a heterogeneous signcryption system that enabled drones to communicate with a GS without a bilinear pairing operation. In the scheme proposed by Pan et al. [26], the drones belonged to IBC and the GS to PKI. The proposed scheme safeguarded the identity of drones and enabled the GS to verify batches. Due to its limited processing capabilities, bilinear pairing is computationally costly for drones to complete. In order to overcome these restrictions, we proposed a conditional privacy-preserving heterogeneous signcryption scheme for IoD that leveraged HECC operation, an improved version of ECC with short keys. The proposed method offered the same level of security as existing systems while incurring minimal computational and communication costs.

## 3. Preliminaries

This section provides the preliminaries, which included the network model, elliptic curve cryptography, the basics of the hyperelliptic curve (HEC) as well as the associated difficult problems (i.e., the hyperelliptic curve Diffie–Hellman problem (HECDHP) and the hyperelliptic curve discrete logarithm problem (HECDLP)), and the syntax of the proposed scheme. Table 1 illustrates the notations used in the construction of the proposed scheme.

### 3.1. Network Model

Figure 1 depicts the network model for the proposed scheme, which consisted of three clusters: Drones, the PKGC, and Everything. The Drones were equipped with cameras, inertial measurement units (IMUs), sensors, and a Global Positioning System (GPS) that could be used in a variety of scenarios. When the Drones wanted to communicate with a device in Everything’s cluster, they sent a request to the PKGC along with their encrypted identity and public and private keys. Further, upon the request of a Drone’s device, the PKGC would generate the private and public keys and send them to the Drone’s device in an encrypted format. By using the received private and public messages, a Drone’s device would generate signcryption on some messages and send the signcrypted text to a device belonging to the Everything cluster. After receiving the signed encrypted text, the device joins the cluster and generated its public and private keys before sending a request for certification to the PKGC. When the PKGC received a request, it generated a certificate and sent it to the device that was shared by all devices in the Everything cluster. By using its private key and the Drone’s public key, a device belonging to the Everything cluster could verify a signature and recover a message. Note that all possible nodes such as GSs, APs, mobile phones, and vehicles on the ground could be included in the Everything cluster. Nonetheless, we only took the GSs into consideration in the proposed network model. The GSs could provide Internet access to the Drones. The Drones used 5G and Wi-Fi wireless technology to connect to the GSs. The drones could communicate with the GSs through 5G and with each other via Wi-Fi. Utilizing the best features of both technologies was important to the hybridization process.

### 3.2. Hyperelliptic Curve (HEC) and Difficult Mathematics Problems

In this subsection, we will cover the basics of the hyperelliptic curve (HEC) as well as the difficult problems; i.e., the hyperelliptic curve Diffie–Hellman problem (HECDHP) and the hyperelliptic curve discrete logarithm problem (HECDLP).

Hyperelliptic Curve (HEC): This is a special form of ECC with genus (g) ≥ 2
that employs 80-bit keys and parameters to generate ciphertext and signatures with the same level of security as ECC. A standard equation for *HEC* over a finite field ($f_{n})$ is as follows: $w^{2}+h\left(a\right)w=f\left(a\right)$
*mod*
n*;*
h(a)∈ F(a) represents a polynomial with degree h(a) ≤ (g)
*and*
f(a) ∈ F(a) represents a monic polynomial with degree f(a)≤ 2(g)+1. Here, the central idea is to construct a Jacobian group and pick its generator, known as the devisor.Hyperelliptic Curve Diffie–Hellman Problem (HECDHP): Assuming the primary parameters for the HECDHP are (∝,ν,(Z=∝·ν·P)), the attacker’s goal, with the help of the challenger, is to extract ∝ and ν from Z.Hyperelliptic Curve Discrete Logarithm Problem (HECDLP): Assuming (∝,(Z=∝·P))
are the main parameters for the HECDLP, the attacker’s goal, with the help of the challenger, is to extract ∝ from Z.

### 3.3. Syntax

The syntax of the proposed scheme consisted of the five algorithms listed below:
Setup: When the private key generation center (PKGC) receives Ø
as a security parameter, it sets ${µ}_{PKGC}$ as its private key and $\Upsilon_{PKGC}$ as a public key. Moreover, it makes $\xi_{PKGC}$ a param.IBC Key Generation for
Drone
**:** Here, first Drone computes $(\delta_{Drone}$, $SK_{sec},Drone_{EID})$ and sends ($Drone_{EID},\delta_{Drone}$) to the PKGC through an insecure channel. The PKGC then computes the secret key $SK_{sec},$ $Drone_{RID}$, $\beta_{Drone}$, and $\pi_{1}$. The PKGC also computes the private key for Drone ($PK_{Drone})$ and Ψ. PKGC sends Ψ to Drone in an open network. The Drone can recover $\left(PK_{Drone},\beta_{Drone}\right)\;from$ (Ψ) later.PKI Key Generation for Everything (EVTG): A device that belongs to the *EVTG* can play the role of receiver and sets $\lambda_{EVTG}$
as its private key and computes $\sigma_{EVTG}$ as its public key.Heterogeneous Signcryption (HS): This step is initiated by the Drone
to generate and send ($S_{Drone},\;\chi_{Drone},\;C_{Drone}$) to the *EVTG.*Heterogeneous Unsigncryption (HUS): A device that belong to *EVTG* can play the role of receiver and can verify and decrypt ($S_{Drone},\;\chi_{Drone},\;C_{Drone}$
).

## 4. Construction of the Proposed Scheme

The construction of the proposed scheme included the following steps.

Setup: When the PKGC receives Ø
as a security parameter, it then performs the following steps:
Selects ${µ}_{PKGC}\;$ randomly, where ${µ}_{PKGC}\in f_{n}\;$ and sets it as its private key;Computes $\Upsilon_{PKGC}={µ}_{PKGC}\cdot P$
and sets it as its private key, where P is the devisor on HECC;Chooses hash functions $H_{a1}$ , $H_{a2}$*,* and $H_{a3}$*,* with a 256-bit size;Sets $\xi_{PKGC}=\{H_{a1}$, $H_{a2},H_{a3},\Upsilon_{PKGC},P,f_{n},\mathrm{HEC}\}$ as a param for further processing of the proposed scheme and the PKGC shares it openly.IBC Key Generation for Drone: Here, first Drone selects (${\mathrm{Drone}}_{RID}$ ) as its real identity and selects $\zeta_{Drone}\in f_{n},\;$ computes $\delta_{Drone}=\zeta_{Drone}\cdot P$, $SK_{sec}=\zeta_{Drone}\cdot \Upsilon_{PKGC}$, encrypts $Drone_{RID}$ as $Drone_{EID}=E_{SK_{sec}}\left(Drone_{RID}\right)$, and sends ($Drone_{EID},\delta_{Drone}$) to the PKGC through an insecure channel. When ($Drone_{RID},\delta_{Drone}$) sends to the PKGC, it computes the secret key $SK_{sec}$ as $SK_{sec}=\delta_{Drone}\cdot {µ}_{PKGC}$, recovers $Drone_{RID}$ as $Drone_{RID}=D_{SK_{sec}}\left(Drone_{EID}\right)$, selects $\eta_{Drone}\in f_{n}$, computes $\beta_{Drone}=\eta_{Drone}\cdot P$, and $\pi_{1}=H_{a1}\left(\beta_{Drone},Drone_{RID}\right)$. Then, the PKGC computes the private key for Drone as $PK_{Drone}=\eta_{Drone}+\pi_{1}\cdot {µ}_{PKGC}$ and encrypt ($PK_{Drone},\beta_{Drone}$) as $\Psi =E_{SK_{sec}}\left(PK_{Drone},\beta_{Drone}\right)$. The PKGC sends Ψ to Drone in an open network, then Drone can recover $\left(PK_{Drone},\beta_{Drone}\right)$ as $\left(PK_{Drone},\beta_{Drone}\right)=D_{SK_{sec}}\left(\Psi \right)$.PKI Key Generation for Everything (EVTG): A device that belongs to the EVTG plays the role of receiver, selects $\lambda_{EVTG}\in f_{n}$, and computes $\sigma_{EVTG}=\lambda_{EVTG}\cdot P$.Heterogeneous Signcryption (HS): This step will be initiated by the Drone
to generates HS using the following steps:
It selects $\rho_{Drone}\in f_{n}$ at random and computes $\chi_{Drone}=\rho_{Drone}\cdot P$;Computes $K=\rho_{Drone}\cdot \sigma_{EVTG}$ and $k=H_{a2}\left(K,\;\chi_{Drone}\right)$;Computes $C_{Drone}=E_{k}\left(m,Drone_{EID}\right)$ and $\pi_{2}=H_{a3}\left(m,\;\chi_{Drone},Drone_{EID}\right)$;Computes $S_{Drone}=\rho_{Drone}+\pi_{2}\cdot PK_{Drone}$ and sends ($S_{Drone},\chi_{Drone},C_{Drone}$) to the EVTG.Heterogeneous Unsigncryption (HUS): A device that belongs to the EVTG plays the role of receiver and can generate HUS using the following steps;Computes $K=\chi_{MUAV}\cdot \lambda_{EVTG}$ and $k=H_{a2}\left(K,\;\chi_{Drone}\right)$;Computes $\left(m,Drone_{EID}\right)=D_{k}\left(C_{Drone}\right)$ and compares if $S_{Drone}\cdot P=\chi_{Drone}+\pi_{2}\left(\sigma_{EVTG}+\pi_{1}\cdot \Upsilon_{PKGC}\right)$ satisfies, where $\pi_{2}=H_{a3}\left(m,\;\chi_{Drone},Drone_{EID}\right)$ and $\pi_{1}=H_{a1}\left(\beta_{Drone},Drone_{RID}\right)$.

### Correctness

The *EVTG* that plays the role of receiver can generate the secret key (K) for decryption as follows:$$K=\chi_{Drone}\cdot \lambda_{EVTG}=\rho_{Drone}\cdot P\;\cdot \lambda_{EVTG}=\rho_{Drone}\;\cdot \lambda_{EVTG}\cdot P=\rho_{Drone}\;\cdot \sigma_{EVTG}$$
hence proved.

In addition, EVTG verifies the signature $S_{MUAV}=\rho_{MUAV}+\pi_{2}\cdot PK_{MUAV}$ as follows:
$$\begin{array}{ll} S_{MUAV}\cdot P & =\chi_{Drone}+\pi_{2}\left(\beta_{Drone}+\pi_{1}\cdot \Upsilon_{PKGC}\right)=S_{Drone}\cdot P\\ & \;\;\;\;\;\;\;\;\;\;\;=\left(\rho_{Drone}+\pi_{2}\cdot PK_{Drone}\right)\cdot P=\left(\rho_{Drone}\cdot P+\pi_{2}\cdot P(PK_{Drone}\right))\\ & \;\;\;\;\;\;\;\;\;\;\;=\left(\rho_{Drone}\cdot P+\pi_{2}\left(\eta_{Drone}+\pi_{1}\cdot \;{µ}_{PKGC}\right)\cdot P\right)\\ & \;\;\;\;\;\;\;\;\;\;\;=\left(\rho_{Drone}\cdot P+\pi_{2}\left(\eta_{Drone}\cdot P+\pi_{1}\cdot \;{µ}_{PKGC}\cdot P\right)\right)\;)\\ & \;\;\;\;\;\;\;\;\;\;\;=\left(\;\chi_{Drone}+\pi_{2}\left(\beta_{Drone}+\pi_{1}\cdot \;\Upsilon_{PKGC}\right)\right) \end{array}$$
hence proved.

## 5. Security Models

In this section, we define the role of two adversary (outsider adversary ($OUT_{ADV}$) and forger ($OUT_{FRGR}$)) that could break the proposed scheme security aspects such as confidentiality and forgeability. The following two games defined the basic preliminaries for confidentiality security defenses against $OUT_{ADV}$ and unforgeability against $OUT_{FRGR}$.

Game 1← Confidentiality: The proposed IBC-PKI-HS Indistinguishability Against Adaptive Chosen Cyphertext Attacks *(IND-CCATK- IBC-PKI-HS)* under HECDHP; whether the outsider adversary $\left(OUT_{ADV}\right)$ with negligible advantages (∂) can solve HECDHP using a challenger $C_{HS}$ was a subroutine.

***Setup:*** By using Ø as a security parameter, the $C_{HS}$ secret key is ${µ}_{PKGC}$, $\xi_{PKGC}$, and the param $\xi_{PKGC}$ is sent to $OUT_{ADV}$.

Phase 1: $OUT_{ADV}$ can make the following queries with $C_{HS}:$

$QRY_{H_{ai}}Query:$ $C_{HS}$ set the lists ($L_{H_{ai}}$) with some initial values. Upon the query request from $OUT_{ADV}$, $C_{HS}$ checks the corresponding value in $L_{H_{ai}}$; if it exists, then $C_{HS}$ sends the requested value to $OUT_{ADV}$. Otherwise, $C_{HS}$ picks the requested value randomly, updates $L_{H_{ai}},$ and sends it to $OUT_{ADV}$.

Public Key Query ($QRY_{PBK\;}$): Here, we consider two cases for user key generation when $OUT_{ADV}$ sends a request for a public key.

Case 1: Upon request of $OUT_{ADV}$ for the keys, which belong to identity-based cryptography, $C_{HS}$ transmits $\beta_{i}$ to $OUT_{ADV}$.

Case 2: Upon request of $OUT_{ADV}$ for the keys, which belong to public key infrastructure-based cryptography, $C_{HS}$ transmits $\sigma_{i}$ to $OUT_{ADV}$.

Private Key Query ($QRY_{PRK\;}$): Here, we consider two cases for private key generation when $OUT_{ADV}$ requests a private key.

Case 1: Upon request of $OUT_{ADV}$ for the private key, which belongs to identity-based cryptography, $C_{HS}$ transmits $PK_{i}\;$ to $OUT_{ADV}$.

Case 2: Upon request of $OUT_{ADV}$ for the private key, which belongs to public key infrastructure-based cryptography, $C_{HS}$ transmits $\lambda_{i}$ to $OUT_{ADV}.$

Heterogeneous Signcryption Query ($QRY_{HS\;}$): Upon request of $OUT_{ADV}$ for the heterogeneous signcryption oracle, $C_{HS}$ transmits ($S_{i},\chi_{i},C_{i}$) to $OUT_{ADV}$.

Heterogeneous Unsigncryption Query ($QRY_{HUS\;}):$ Upon request of $OUT_{ADV}$ for the heterogeneous signcryption oracle, $C_{HS}$ either returns with plaintext or confirms ($S_{i},\;\;\chi_{i},C_{i}$) is invalid.

Challenge: $OUT_{ADV}$ sends the triple ($m_{1}$, $m_{1},Drone_{RID},\;ID_{\mathrm{EVTG}}$) to $C_{HS}$, which will respond with ($S_{i}{}^{*},\chi_{i}{}^{*},C_{i}{}^{*}$) to $OUT_{ADV}$.

Phase 2: $OUT_{ADV}$ represents the same nature of queries as made in Phase 1 except for using $QRY_{PRK\;}$ for $MUAV_{RID}$. In addition, $OUT_{ADV}$ will not generate a request for plaintext that is related to ($S_{i}{}^{*},\chi_{i}{}^{*},C_{i}{}^{*}$).

Guess: $OUT_{ADV}$ produces $\tau^{/}$. If $\tau =\tau^{/},$ $C_{HS}$ returns a true result; otherwise it returns a false result.

Game 2← Unforgeability (UU-ACMA- IBC-PKI-HS): The proposed IBC-PKI-HS Unforgeable Under Adaptive Chosen Message Attacks *(UU-ACMA- IBC-PKI-HS)* under HECDLP; whether the Forger $\left(OUT_{FRGR}\right)$ with advantages (∂) can solve the HECDLP using a challenger $C_{HS}$ is a subroutine.

Setup: By using Ø as a security parameter, the $C_{HS}$ secret key is ${µ}_{PKGC}$, $\xi_{PKGC}$, and the param $\xi_{PKGC}$ is sent to $OUT_{FRGR}$.

Phase 1: $OUT_{FRGR}$ can make the following queries with $C_{HS}$:

$QRY_{H_{ai}}Query:$ $C_{HS}$ set the lists ($L_{H_{ai}}$) with some initial values. Upon the query request from $OUT_{ADV}$, $C_{HS}$ checks the corresponding value in $L_{H_{ai}}$; if it exists, then $C_{HS}$ sends the requested value to $OUT_{FRGR}$. Otherwise, $C_{HS}$ picks the requested value randomly, updates $L_{H_{ai}},$ and sends it to $OUT_{FRGR}$.

Public Key Query ($QRY_{PBK\;}$): Here, we consider two cases for user key generation when $OUT_{FRGR}$ sends a request for a public key.

Case 1: Upon request of $OUT_{FRGR}$ for the keys, which belong to identity-based cryptography, $C_{HS}$ transmits $\beta_{i}$ to $OUT_{FRGR}$.

Case 2: Upon request of $OUT_{FRGR}$ for the keys that belong to public key infrastructure-based cryptography, $C_{HS}$ transmits $\sigma_{i}$ to $OUT_{FRGR}$.

Private Key Query ($QRY_{PRK\;}$): Here, we consider two cases for private key generation when $OUT_{FRGR}$ requests for a private key.

Case 1: Upon request of $OUT_{FRGR}$ for the private key, which belongs to identity-based cryptography, $C_{HS}$ transmits $PK_{i}\;$ to $OUT_{FRGR}$.

Case 2: Upon request of $OUT_{FRGR}$ for the private key, which belongs to public key infrastructure-based cryptography, $C_{HS}$ transmits $\lambda_{i}$ to $OUT_{FRGR}.$

Heterogeneous Signcryption Query ($QRY_{HS\;})$: Upon request of $OUT_{FRGR}$ for the heterogeneous signcryption oracle, $C_{HS}$ transmits ($S_{i},\chi_{i},C_{i}$) to $OUT_{FRGR}$.

**Forgery:** $OUT_{FRGR}$ can generate a forge signcryption ($S_{i}{}^{*},\chi_{i}{}^{*},C_{i}{}^{*}$) if the following steps are successfully completed:

Step 1: $QRY_{PRK\;}C_{HS}$ succeeds.Step 2: $QRY_{HUS\;}C_{HS}$ succeeds.Step 3: All the queries are successful in target identity.

## 6. Security Analysis

In this part, we demonstrate that the proposed scheme was secure against confidentiality and unforgeability breaches under the random oracle model (ROM).

**Theorem 1.** *Confidentiality (IND-CCATK- IBC-PKI-HS)*.

The proposed IBC-PKI-HS Indistinguishability Against Adaptive Chosen Cyphertext Attacks *(IND-CCATK- IBC-PKI-HS)* was under the HECDHP. Whether the outsider adversary $\left(OUT_{ADV}\right)$ with advantages (∂) could solve the HECDHP using a challenger $C_{HS}$ was a subroutine. The following is the success advantage of $C_{HS}$ in which it can solve HECDHP for $OUT_{ADV}$:$$\mathrm{Prob}\left(C_{HS}\;success\right)=\left(1-\frac{QRY_{PRK}}{QRY_{PBK}}\right)\left(1-\frac{1}{2^{Ø\;}}\right)\left(\frac{1}{QRY_{PBK}-QRY_{PRK}}\right)\partial ,\;$$
where $QRY_{PBK}$ is the public key query and $QRY_{PRK}$ is the private key query.

**Proof:** Suppose (∝,ν,(Z=∝·ν·P))
is the HECDHP: the task of $OUT_{ADV}$ with the help of $C_{HS}$ is to extract ∝ and ν from Z by using the following steps:Setup: By using Ø as a security parameter, $C_{HS}$ secret key as ${µ}_{PKGC}$, public key $\Upsilon_{PKGC}$, $\xi_{PKGC}$, and send $\Upsilon_{PKGC}$ and $\xi_{PKGC}$ to $OUT_{ADV}$.Phase 1: $OUT_{ADV}$ can make the following queries with $C_{HS}.$$QRY_{H_{a1}}Query:$ $C_{HS}$ sets a list ($L_{H_{a1}}$) with tuple $\left(\beta_{i},Drone_{RIDi},\pi_{1i}\right)$. Upon the query request from $OUT_{ADV}$, $C_{HS}$ checks the value $\pi_{1i}$ in $L_{H_{a1}}$; if $\pi_{1i}$ exists, then $C_{HS}$ sends $\pi_{1i}$ to $OUT_{ADV}$. Otherwise, $C_{HS}$ picks the value $\pi_{1i}$ randomly, updates $L_{H_{a1}},$ and sends $\pi_{1i}$ to $OUT_{ADV}$.$QRY_{H_{a2}}Query:$ $C_{HS}$ sets a list ($L_{H_{a2}}$) with tuple $\left(K_{i},\;\chi_{i},k_{i}\right)$. Upon the query request from $OUT_{ADV}$, $C_{HS}$ checks the value $k_{i}$ in $L_{H_{a2}}$; if $k_{i}\;$ exists, then $C_{HS}$ sends $k_{i}$ to $OUT_{ADV}$. Otherwise, $C_{HS}$ picks the value $k_{i}$ randomly, updates $L_{H_{a2}},$ and sends $k_{i}$ to $OUT_{ADV}$.$QRY_{H_{a3}}Query:$ $C_{HS}$ sets a list ($L_{H_{a3}}$) with tuple $\left(m_{i},\;\chi_{i},Drone_{EIDi}\right)$. Upon the query request from $OUT_{ADV}$, $C_{HS}$ checks the value $\pi_{2i}$ in $L_{H_{a3}}$; if $\pi_{2i}$ exists, then $C_{HS}$ sends $\pi_{2i}$ to $OUT_{ADV}$. Otherwise, $C_{HS}$ picks the value $\pi_{2i}$ randomly, updates $L_{H_{a3}},$ and sends $\pi_{2i}$ to $OUT_{ADV}$.***Public Key Query (***$QRY_{PBK\;}$***):*** Here, we consider two cases for user key generation when $OUT_{ADV}$ asks for this query.Case 1: Upon request of $OUT_{ADV}$ for the keys that belong to identity-based cryptography, $C_{HS}$ checks the tuple ($\beta_{i},Drone_{RIDi}$) in list $L_{pbk}$; if it is found, $C_{HS}$ transmits $\beta_{i}$ to $OUT_{ADV}$. Otherwise, at the j^th^ query, $C_{HS}$ computes $\beta_{i}=\propto \cdot P$. Further, $C_{HS}$ checks if i≠ j, then computes $\beta_{i}=\eta_{i}\cdot P$, where $\eta_{i}$ is randomly selected number. Then, $C_{HS}$ updates the list $L_{pbk}$ and sends $\beta_{i}$ to $OUT_{ADV}.$*Case 2:* Upon request of $OUT_{ADV}$ for the keys that belong to public key infrastructure-based cryptography, $C_{HS}$ checks the tuple ($\sigma_{i},ID_{i}$) in list $L_{cuk}$; if it is found, $C_{HS}$ transmits $\sigma_{i}$ to $OUT_{ADV}$. Otherwise, $C_{HS}$ computes $\sigma_{i}=\propto \cdot P$, updates the list $L_{cuk}$, and sends $\sigma_{i}$ to $OUT_{ADV}.$Private Key Query ($QRY_{PRK\;}$): Here, we consider two cases for private key generation when $OUT_{ADV}\;\mathrm{asks}\;\mathrm{for}\;\mathrm{this}\;\mathrm{query}.$ Case 1: Upon request of $OUT_{ADV}$ for the private key that belongs to identity-based cryptography, $C_{HS}$ checks if $Drone_{RIDi}=Drone_{target}$, then aborts this game. Otherwise, it finds the tuple ($\beta_{i},Drone_{RIDi}$, $PK_{i}$) in list $L_{prk}$ and transmits $PK_{i}$ to $OUT_{ADV}$.Case 2: Upon request of $OUT_{ADV}$ for the private key that belongs to public key infrastructure-based cryptography, $C_{HS}$ checks if $ID_{i}=ID_{target}$, then aborts this game. Otherwise, it finds the tuple ($\sigma_{i},ID_{i}$, $\lambda_{i}$) in the list $L_{prk}$*,* and transmits $\lambda_{i}$ to $OUT_{ADV}.$Heterogeneous Signcryption Query ($QRY_{HS\;})$: Upon request of $OUT_{ADV}$ for the heterogeneous signcryption oracle with tuple ($Drone_{RID},m,\;ID_{\mathrm{EVTG}}$), where RID is the identity of Drone, m is the plaintext, and $ID_{\mathrm{EVTG}}$ is the identity of the EVTG. Then, $C_{HS}$ performs the following steps when $Drone_{RIDi}\neq ID_{target}$:Selects $\rho_{i},\pi_{2i}\in f_{n}$ at random and computes $\;\chi_{i}=\rho_{i}\cdot P$;Computes $K=\rho_{i}\cdot \sigma_{i}$ and extracts $k_{i}$ from $L_{H_{a2}}$;Computes $C_{i}=E_{k_{i}}\left(m,Drone_{EID}\right)$ and selects $S_{i}\in f_{n}$;Sends ($S_{i},\chi_{i},C_{i}$ ) to $OUT_{ADV}$.Heterogeneous Unsigncryption Query ($QRY_{HUS\;}):$ Upon request of $OUT_{ADV}$ for the heterogeneous signcryption oracle, $C_{HS}$ checks if $ID_{\mathrm{EVTG}}\neq ID_{target}$ and performs the following steps:Computes $K=\chi_{Drone}\cdot \lambda_{EVTG}$ and $\pi_{2}=H_{a2}\left(K,\;\chi_{Drone}\right)$;Computes $\left(m,Drone_{EID}\right)=D_{k}\left(C_{Drone}\right)$ and compares to determine if $S_{Drone}\cdot P=\chi_{Drone}+\pi_{2}\left(\sigma_{EVTG}+\pi_{1}\cdot \Upsilon_{PKGC}\right)$ satisfied, where $\pi_{2}=H_{a3}\left(m,\;\chi_{Drone},Drone_{EID}\right)$ and $\pi_{1}=H_{a1}\left(\beta_{Drone},Drone_{RID}\right)$.Otherwise, $C_{HS}$ confirms that ($S_{i},\;\chi_{i},\;C_{i}$) is invalid.Challenge: $OUT_{ADV}$ sends the triple ($m_{1}$, $m_{1},Drone_{RID},\;ID_{\mathrm{EVTG}}$) to $C_{HS}$, where ($m_{1}$, $m_{1}$) are the two messages with equal lengths but different contents, and ($Drone_{RID},\;ID_{\mathrm{EVTG}}$) is the identity of Drone and the EVTG. After this, $C_{HS}$ checks whether $ID_{\mathrm{EVTG}}\neq ID_{target}$ and performs the following steps:Selects τ∈ {0, 1} and chooses $\rho_{i}$,$\nu ,k\in f_{n}$;Computes $\;\chi_{i}=\nu \cdot P$ and $K=\rho_{i}+Z$;Computes $C_{i}=E_{k}\left(m\right)$ and $\pi_{2}=H_{a3}\left(m,\;\chi_{Drone},Drone_{EID}\right)$;Computes $S_{i}=\rho_{i}+\nu +\pi_{2}\cdot PK_{Drone}$ and sends ($S_{i}{}^{*},\chi_{i}{}^{*},C_{i}{}^{*}$) to $OUT_{ADV}$.Phase 2. $OUT_{ADV}$ uses the same nature of queries as made in Phase 1 except using $QRY_{PRK\;}$ for $Drone_{RID}$. In addition, $OUT_{ADV}$ will not generate a request for plaintext that is related to ($S_{i}{}^{*},\chi_{i}{}^{*},C_{i}{}^{*}$).Guess: $OUT_{ADV}$ produced $\tau^{/}$. If $\tau =\tau^{/},$ $C_{HS}$ returns a true result; otherwise it returns a false result. If Z=∝·ν·P, then ($S_{i}{}^{*},\chi_{i}{}^{*},C_{i}{}^{*}$) is not valid.Probability Analysis: The following are some events in which $C_{HS}$ will not fail:
Event 1 ($\ell_{1}$): $QRY_{PRK\;}C_{HS}$ succeeds, and the probability as $\left(1-\frac{QRY_{PRK}}{QRY_{PBK}}\right)$
Event 2 ($\ell_{2}$): $QRY_{HUS\;}C_{HS}$ succeeds, and the probability as $\left(1-\frac{1}{2^{Ø}}\right)$
Event 3 ($\ell_{3}$): The challenge phase succeeds, and the probability is $\left(\frac{1}{QRY_{PBK}-QRY_{PRK}}\right)\partial$
So, the following results can be obtained:$$\mathrm{Prob}\left(\ell_{1}\right)=\left(1-\frac{QRY_{PRK}}{QRY_{PBK}}\right),\;\mathrm{Prob}\left(\ell_{2}\right)=\left(1-\frac{1}{2^{Ø}}\right),\;\mathrm{Prob}\left(\ell_{3}\right)=\left(\frac{1}{QRY_{PBK}-QRY_{PRK}}\right)\partial$$
$$\mathrm{Prob}\left(C_{HS}\;success\right)=\mathrm{Prob}(\ell_{1}\wedge \ell_{2}\wedge \ell_{3})=\mathrm{Prob}\left(\ell_{1}\right)\cdot \mathrm{Prob}\left(\ell_{2}\right)\cdot \mathrm{Prob}\left(\ell_{3}\right)$$
$$\mathrm{Prob}\left(C_{HS}\;success\right)=\left(1-\frac{QRY_{PRK}}{QRY_{PBK}}\right)\left(1-\frac{1}{2^{Ø\;}}\right)\left(\frac{1}{QRY_{PBK}-QRY_{PRK}}\right)\partial$$□


**Theorem 2.** 
*Unforgeability (UU-ACMA- IBC-PKI-HS).*


The proposed IBC-PKI-HS Unforgeable Under Adaptive Chosen Message Attacks *(UU-ACMA- IBC-PKI-HS)* was under the HECDLP. Whether the Forger $\left(OUT_{FRGR}\right)$ with advantages (∂) could solve the HECDLP using a challenger $C_{HS}$ was a subroutine. The following is the success advantage of $C_{HS}$ in which it could solve the HECDLP for $OUT_{FRGR}$:$$\mathrm{Prob}\left(C_{HS}\;success\right)=\left(1-\frac{QRY_{PRK}}{QRY_{PBK}}\right)\left(1-\frac{1}{2^{Ø\;}}\right)\left(\frac{1}{QRY_{PBK}-QRY_{PRK}}\right)\partial ,\;$$
where $QRY_{PBK}$ is the public key query and $QRY_{PRK}$ is the private key query.

**Proof:** Suppose (∝,(Z=∝·P)) is the HECDLP: the task of $OUT_{FRGR}$ with the help of $C_{HS}$ is to extract ∝ from Z by using the following steps.Setup: By using Ø as a security parameter, $C_{HS}$ secret key as ${µ}_{PKGC}$, public key $\Upsilon_{PKGC}$, $\xi_{PKGC}$, and send $\Upsilon_{PKGC}$ and $\xi_{PKGC}$ to $OUT_{FRGR}$.***Queries****:*$OUT_{FRGR}$ can make the same queries with $C_{HS}$ as used in the confidentiality *Game.*Forgery: $OUT_{FRGR}$ can generate a forge signcryption ($S_{i}{}^{*},\chi_{i}{}^{*},C_{i}{}^{*}$) if the following computations are successfully done:The $C_{HS}$ must be the original value for $\rho_{MUAV}$; this is only possible if it obtains the solution for Z=∝·PIn addition, $C_{HS}$ must be the original value for $PK_{MUAV}$; this is only possible if it obtains the solution for Z=∝·P during the *public key query (*$QRY_{PBK\;}$*) and* the *private key query (*$QRY_{PRK\;}$*)* or it can access the exact value from list $L_{prk}$.It can also extract the exact value as used in the heterogeneous signcryption algorithm for $\pi_{2}$ from a list ($L_{H_{a3}}$).It can extract the exact value as used in the heterogeneous signcryption algorithm for k from a list ($L_{H_{a2}}$).Probability Analysis: The following are some events in which $C_{HS}$ will not fail:
Event 1 ($\ell_{1}$): $QRY_{PRK\;}C_{HS}$ succeeds, and the probability is $\left(1-\frac{QRY_{PRK}}{QRY_{PBK}}\right)$
Event 2 ($\ell_{2}$): $QRY_{HUS\;}C_{HS}$ succeeds, and the probability is $\left(1-\frac{1}{2^{Ø}}\right)$
Event 3 ($\ell_{3}$): The challenge phase succeeds, and the probability is $\left(\frac{1}{QRY_{PBK}-QRY_{PRK}}\right)\partial$
So, the following results can be obtained:$$\mathrm{Prob}\left(\ell_{1}\right)=\left(1-\frac{QRY_{PRK}}{QRY_{PBK}}\right),\;\mathrm{Prob}\left(\ell_{2}\right)=\left(1-\frac{1}{2^{Ø}}\right),\;\mathrm{Prob}\left(\ell_{3}\right)=\left(\frac{1}{QRY_{PBK}-QRY_{PRK}}\right)\partial$$
$$\mathrm{Prob}\left(C_{HS}\;success\right)=\mathrm{Prob}(\ell_{1}\wedge \ell_{2}\wedge \ell_{3})=\mathrm{Prob}\left(\ell_{1}\right)\cdot \mathrm{Prob}\left(\ell_{2}\right)\cdot \mathrm{Prob}\left(\ell_{3}\right)$$
$$\mathrm{Prob}\left(C_{HS}\;success\right)=\left(1-\frac{QRY_{PRK}}{QRY_{PBK}}\right)\left(1-\frac{1}{2^{Ø\;}}\right)\left(\frac{1}{QRY_{PBK}-QRY_{PRK}}\right)\partial$$□

**Theorem 3.** *Sender Anonymity*.

The proposed IBC-PKI-HS resists against the disclosure of the sender’s identity under the hardiness of the HECDLP.

**Proof:** In the proposed scheme, the Drone device selects (${\mathrm{Drone}}_{\mathrm{RID}}$) as its real identity, selects $\zeta_{Drone}\in f_{n},\;$**computes**
$\delta_{Drone}=\zeta_{Drone}\cdot P$, $SK_{sec}=\zeta_{Drone}\cdot \Upsilon_{PKGC}$, encrypts $Drone_{RID}$ as $Drone_{EID}=E_{SK_{sec}}\left(Drone_{RID}\right)$, and sends ($Drone_{EID},\delta_{Drone}$) to the PKGC through an insecure channel. When ($Drone_{RID},\delta_{Drone}$) sends to the PKGC, it computes the secret key $SK_{sec}$ as $SK_{sec}=\delta_{Drone}\cdot {µ}_{PKGC}$, recovers $Drone_{RID}$ as $Drone_{RID}=D_{SK_{sec}}\left(Drone_{EID}\right)$, selects $\eta_{Drone}\in f_{n}$, computes $\beta_{Drone}=\eta_{Drone}\cdot P$, and $\pi_{1}=H_{a1}\left(\beta_{Drone},Drone_{RID}\right)$. Here, the Drone device acts as a sender and if $OUT_{ADV}$ wants the real identity ${\mathrm{Drone}}_{RID}$ of Drones, then it must reveal the secret key $SK_{sec}=\zeta_{Drone}\cdot \Upsilon_{PKGC}$. To do so, it needs the value $\zeta_{Drone}$ from $\delta_{Drone}=\zeta_{Drone}\cdot P$ that is equal to solve the HECDLP, which is infeasible for $OUT_{ADV}$.□

**Theorem 4.** *Receiver Anonymity*.

The proposed IBC-PKI-HS resists against the disclosure of the receiver’s identity.

**Proof:** We did not use the receiver identity in any communication process, so our proposed scheme provided receiver anonymity.□

## 7. Performance Comparison

This section compares the performance of the proposed scheme with the relevant existing counterparts proposed by Ali et al. [21], Jin et al. [23], Ting et al. [24], Ali et al. [25], and Pan et al. [26] based on the security properties, computation cost, and communication cost.

### 7.1. Security Properties Comparison

In this section, we made a comparison regarding the security properties between the proposed scheme and those of Ali et al. [21], Jin et al. [23], Ting et al. [24], Ali et al. [25], and Pan et al. [26]. The comparison was made using Table 2, in which we included the security properties such as the confidentiality, unforgeability, sender’s anonymity, receiver’s anonymity, and needing a secure channel. Further, if a scheme obeyed the security properties, we indicated “Yes” or vice versa. Moreover, if the scheme security analysis section did not include an explanation of the security properties, we indicated “Not Mentioned.” The proposed scheme provided all the security requirements that are used in Table 1, while the schemes used in Ali et al. [21], Jin et al. [23], Ting et al. [24], Ali et al. [25], and Pan et al. [26] did not provide a secure-channel-free environment for the distribution of a private key between a user and the PKGC. In addition, the schemes used in Ali et al. [21], Jin et al. [23], Ting et al. [24], Ali et al. [25], and Pan et al. [26] did not explain the security requirements for sender and receiver anonymity.

### 7.2. Computation Costs

The computation costs represented the operational expenses consumed by each user during the proposed scheme and the existing comparable schemes proposed by Ali et al. [21], Jin et al. [23], Ting et al. [24], Ali et al. [25], and Pan et al. [26]. Table 3 lists the key operations of the proposed scheme, which included bilinear-pairing-based multiplication (BPM), exponentials (EX), elliptic curve point multiplication (EM), hyperelliptic curve point multiplication (HEM), and bilinear pairing operations (PR).

Table 4 contains the operating expenses as measured in milliseconds (ms) for the proposed scheme as well as those of Ali et al. [21], Jin et al. [23], Ting et al. [24], Ali et al. [25], and Pan et al. [26]. The time requirements for a single BPM were 4.31 ms; EX, 1.25 ms; EM, 0.97 ms; HEM, 0.48 ms; and PR, 14.90. The Multi-Precision Integer and Rational Arithmetic C Library (MIRACL) [27] was used to assess the performance of the proposed scheme by testing the runtime of the core cryptographic operations up to 1000 times. Observations were made on a workstation with the following specifications: 8 GB RAM and the Windows 7 Home Basic 64-bit operating system [28]. As seen in Figure 2, the proposed scheme had a lower computation cost than the schemes proposed by Ali et al. [21], Jin et al. [23], Ting et al. [24], Ali et al. [25], and Pan et al. [26].

### 7.3. Communication Costs

In this subsection, the proposed scheme is compared to the existing schemes; namely, those proposed by Ali et al. [21], Jin et al. [23], Ting et al. [24], Ali et al. [25], and Pan et al. [26] in terms of the communication costs. We listed the communication costs incurred based on the elliptic curve parameter size (*|ECC q|*), bilinear pairing parameter size (*|BP G|*), and a message size (*|m|*) for the proposed scheme and those of Ali et al. [21], Jin et al. [23], Ting et al. [24], Ali et al. [25], and Pan et al. [26]. We selected the bit values for the bilinear pairing group size (|G|=1024 bits), elliptic curve parameter size (|q| = 160 bits), hyperelliptic curve parameter size (|n| = 80 bits), and message size (|m| = 1024 bits).

In addition, the communication cost analysis between the schemes of Ali et al. [21], Jin et al. [23], Ting et al. [24], Ali et al. [25], Pan et al. [26] and the proposed scheme are provided in Table 5. As seen in Figure 3, the proposed scheme had a lower communication cost than the schemes proposed by Ali et al. [21], Jin et al. [23], Ting et al. [24], Ali et al. [25], and Pan et al. [26].

## 8. Conclusions

In this article, we proposed a heterogeneous signcryption scheme with an option for conditional privacy. In the proposed scheme, drones employed identity-based cryptography (IBC) while the ground station (GS) used the public key infrastructure (PKI). The proposed scheme was built on the hyperelliptic curve cryptosystem (HECC), and its security robustness was assessed using the random oracle model (ROM). In addition, we introduced a new idea in IBC for the proposed method in which the PKGC communicated the private key to drones in an encrypted format that did not require a secure channel. A complete investigation of the ROM’s security revealed that the proposed scheme was resistant to a variety of threats. In terms of the computation and communication costs, when comparing the proposed scheme to comparable schemes described by Ali et al. [21], Jin et al. [23], Ting et al. [24], Ali et al. [25], and Pan et al. [26], the results indicated that the proposed scheme was more cost-effective than the existing options in terms of the computation and communication costs. In addition, the findings indicated that the proposed scheme was suitable for IoD systems due to the algorithm’s functionality and decreased computation and communication costs.

In future work, we intend to improve the proposed scheme so that it provides digital signatures and encryption not only simultaneously but also independently as based on application needs. In addition, we want to use the Automated Validation of Internet Security Protocols and Applications (AVISPA) tool to double-check the security toughness of the proposed scheme.

## Figures and Tables

**Figure 1 sensors-23-01063-f001:**
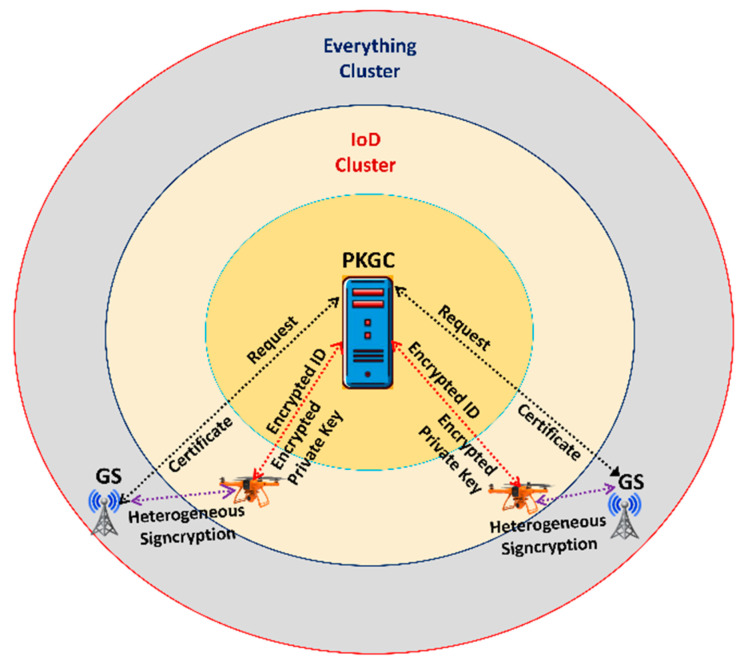
Network model of the proposed scheme.

**Figure 2 sensors-23-01063-f002:**
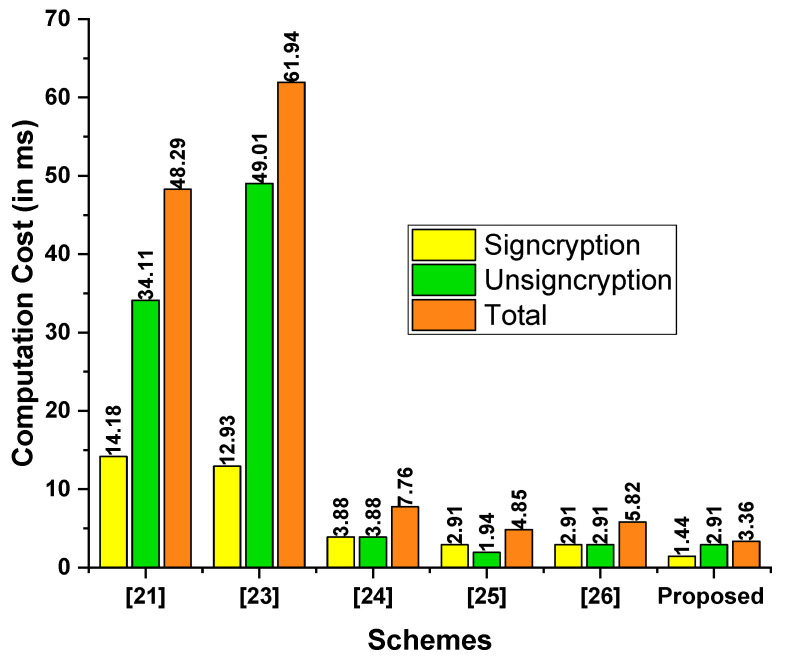
Comparison of computation costs (in ms).

**Figure 3 sensors-23-01063-f003:**
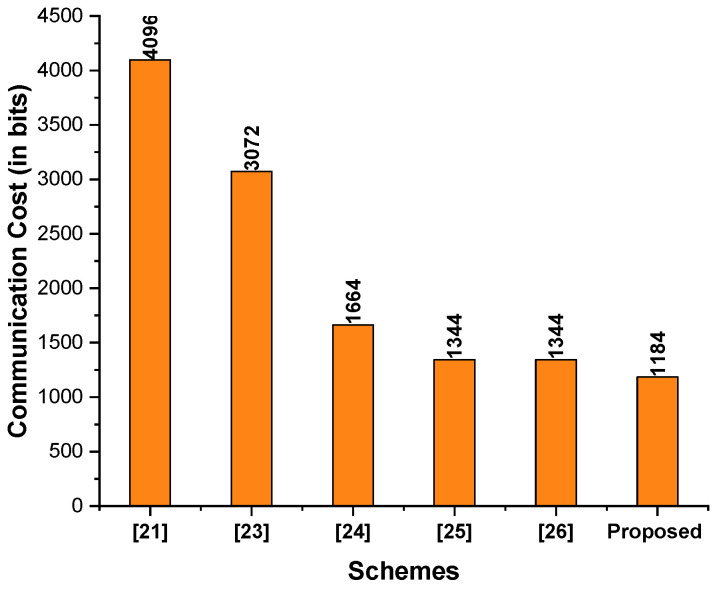
Comparison of communication costs (in bits).

**Table 1 sensors-23-01063-t001:** Notation table.

S.No	Notation	Descriptions
1	PKGC	The private key generation center
2	Ø	A security parameter of HEC with a size of 80 bits
3	${µ}_{PKGC}$	The private key of PKGC
4	$\Upsilon_{PKGC}$	The public key of PKGC
5	$f_{n}$	A finite field with order q=80 bits
6	∈	Belongs to symbol
7	HEC	Genus 2 hyperelliptic curve
8	P	Devisor of genus 2
9	$H_{a1}$, $H_{a2},H_{a3}$	Hash functions with sizes of 256 bits
10	${\mathrm{Drone}}_{RID}$	The real identity of the drone
11	${\mathrm{Drone}}_{EID}$	The encrypted identity of the drone
12	$E_{SK_{sec}}$	The encryption function, which was used to encrypt the real identity of the Drone
13	$D_{SK_{sec}}$	The decryption function, which was used to recover the real identity of the *Drone*
14	$SK_{sec}$	The secret key, which was used to encrypt and decrypt the messages between the *Drone* and the PKGC
15	$PK_{Drone}$	The private key of the *Drone*
16	$\lambda_{EVTG}$	The private key of the device that belonged to *EVTG*
17	$\sigma_{EVTG}$	The public key of the device that belonged to *EVTG*
18	k	The secret key that was used to encrypt and decrypt the messages between the ***Drone*** and the *EVTG*
19	$E_{k}$	The encryption function, which was used to encrypt the message of the *Drone*
20	$D_{k}$	The decryption function, which was used to recover the message of the *Drone*

**Table 2 sensors-23-01063-t002:** Comparison of security properties.

Schemes	Confidentiality	Unforgeability	Sender Anonymity	Receiver Anonymity	Needing Secure Channel
Ali et al. [21]	Yes	Yes	Not Mentioned	Not Mentioned	No
Jin et al. [23]	Yes	Yes	Not Mentioned	Not Mentioned	No
Ting et al. [24]	Yes	Yes	Not Mentioned	Not Mentioned	No
Ali et al. [25]	Yes	Yes	Not Mentioned	Not Mentioned	No
Pan et al. [26]	Yes	Yes	Not Mentioned	Not Mentioned	No
Proposed Scheme	Yes	Yes	Yes	Yes	Yes

**Table 3 sensors-23-01063-t003:** Comparison of computation costs with major operations.

Schemes	Signcryption	Unsigncryption	Total
Ali et al. [21]	3 BPM + 1 EX	1 BPM + 2 PR	4 BPM + 1 EX + 2 PR
Jin et al. [23]	3 BPM	1 BPM + 3 PR	4 BPM + 3 PR
Ting et al. [24]	4 EM	4 EM	8 EM
Ali et al. [25]	3 EM	2 EM	5 EM
Pan et al. [26]	3 EM	3 EM	6 EM
Proposed scheme	3 HEM	4 HEM	7 HEM

**Table 4 sensors-23-01063-t004:** Comparison of computation costs (in ms).

Schemes	Signcryption	Unsigncryption	Total
Ali et al. [21]	14.18	34.11	48.29
Jin et al. [23]	12.93	49.01	61.94
Ting et al. [24]	3.88	3.88	7.76
Ali et al. [25]	2.91	1.94	4.85
Pan et al. [26]	2.91	2.91	5.82
Proposed Scheme	1.44	1.92	3.36

**Table 5 sensors-23-01063-t005:** Comparison of communication costs (in bits).

Schemes	Signcrypted Text Tuple	Signcrypted Text in Bits
Ali et al. [21]	|m|+3|G|	|1024|+3*|1024| = 4096
Jin et al. [23]	|m|+2|G|	|1024|+2*|1024| = 3072
Ting et al. [24]	|m|+4|q|	|1024|+4*|160| = 1664
Ali et al. [25]	|m|+2|q|	|1024|+2*|160| =1344
Pan et al. [26]	|m|+2|q|	|1024|+2*|160| = 1344
Proposed Scheme	|m|+2|n|	|1024|+2*|80| = 1184

## Data Availability

Not applicable.

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
