# Peer review of "An Efficient and Conditional Privacy-Preserving Heterogeneous Signcryption Scheme for the Internet of Drones"

_sensors, 2023, doi:10.3390/s23031063_

Round 1
Reviewer 1 Report
The authors proposed the Conditional privacy Preservice Heterogeneous Signature Scheme for the Internet of Drones, which is interesting and meaningful.
Some problems and suggestions in the manuscript are as follows:
1) The paper believes that five contributions have been made, some of which are just general work of the paper, and do not belong to innovation and contribution. It is suggested to combine them into three.
2) Figure 1. The network model of the proposed scheme is too simple and does not have a solution at the method level. It is suggested to refine and highlight the characteristics of this article.
3) Sections 3.2-3.2 and 4 of the paper are basically descriptions at the definition level of D. It is impossible to judge the importance and specific principle of the work. Moreover, the layout of the typesetting is quite messy, so it is recommended to rearrange the typesetting.
4) The full text basically has no formula and mathematical principles, so what is the theoretical basis of this article and where is the improvement.
Finally, in the experimental stage, there were very few demonstrations of the experimental environment, experimental conditions, experimental platforms, physical objects, etc., and it was impossible to evaluate the rationality of the experiment.
Author Response
Dear Reviewer,
Please find our response letter in the attached document.
Thank you!

Reviewer 2 Report
1. In Section 5, two games which define the basics preliminaries for confidentiality security you mentioned, but I can only find one from it.
2. In Security Analysis, proposed scheme needs to add something to compare with the relevant existing counterparts.
3. The content of Table 2 can be replaced by pictures, which is more intuitive.
4. Part of the contents repeated throughout the paper.
5. Typography is not the most important thing we are concerned about. But there are too many problems that affect the continuity of reading.
Author Response

(The authors gave the same response as above.)

Reviewer 3 Report
The paper proposes a security schema for the Internet of Drones with the emphasis on the efficiency in terms of computation and communication costs, which is important due to limitations of hardware performance of drones. The general idea is good and the contribution clear, but I suggest restructuring the paper to be more interesting to read and easier to follow. Most of the paper consists of a list of mathematical formulas and abbreviations, which makes it tedious to read and follow. Adding some real-world examples would help. Pseudocode for the proposed algorithms may improve clarity. Table 1 with notations should be put before first mentioning of the notations. Most of the paper is written as bullets and numbering with inconsistent formatting and restructuring is suggested.
Security models are two case studies that should be described in more detail. They are presented here as listing. Maybe some figures and real-world scenarios may be shown thorough these case studies.
I suggest to clearly divide the paper into methods and results parts.
Equations should be numbered and separated from the text. This form of writing where equations and text are mixed is less appropriate.
Furthermore, an explanation how it was implemented and tested is required. Describe what hardware/software was used for the performance analysis and how it was tested compared to other cited approaches. On what platform those values in Tables 2-4 and Figures 2,3 were calculated.
What are the limitations of the study and possible future work?
There are some typos, missing or double blank spaces so I suggest going through the text once more and correct it. E.g., line 295, 297, 301, 303 – which is belong to
Author Response

(The authors gave the same response as above.)

Round 2
Reviewer 1 Report
The authors have addressed all my comments. The paper can be accepted as present form,except for some typesetting.
Author Response
The authors have addressed all my comments. The paper can be accepted as present form except for some typesetting.
Response: We are thankful to the reviewer for extolling our efforts. We are grateful that the reviewer is satisfied with our article and has accepted it. Besides, we have revised the article to correct misspellings and punctuation mistakes, as well as conducted a thorough grammatical check. Additionally, missing and double blank spaces are corrected.
Reviewer 3 Report
Some of the suggested improvements from the previous round of review have been addressed. I suggest further improvements of the paper structure, layout, and mathematical expressions, as mentioned in the previous round of the review.
There are two table 1 and no table 2, so the numbering of the tables should be corrected.
Author Response
Some of the suggested improvements from the previous round of review have been addressed. I suggest further improvements of the paper structure, layout, and mathematical expressions, as mentioned in the previous round of the review.
There are two table 1 and no table 2, so the numbering of the tables should be corrected.
Response: We value the reviewer's comment. We attempted to improve the paper's structure, layout, and mathematical expressions. We have revised the write-up following an extensive grammar check. Attempts have been made to assure spontaneity and lexical beautification. We have done our best to keep the reading and flow of the contents consistent.
Table 1 and Table 2 numbering have been corrected.